# High-Density Lipoprotein Suppresses Neutrophil Extracellular Traps Enhanced by Oxidized Low-Density Lipoprotein or Oxidized Phospholipids

**DOI:** 10.3390/ijms232213992

**Published:** 2022-11-13

**Authors:** Hitomi Ohinata, Takashi Obama, Tomohiko Makiyama, Yuichi Watanabe, Hiroyuki Itabe

**Affiliations:** Division of Biological Chemistry, Department of Pharmaceutical Sciences, Showa University School of Pharmacy, Tokyo 142-8555, Japan

**Keywords:** neutrophil extracellular traps, high-density lipoprotein (HDL), oxidized low-density lipoprotein (oxLDL), oxidized phosphatidylcholine, electronegative low-density lipoprotein

## Abstract

Neutrophil extracellular traps (NETs) are found in patients with various diseases, including cardiovascular diseases. We previously reported that copper-oxidized low-density lipoprotein (oxLDL) promotes NET formation of neutrophils, and that the resulting NETs increase the inflammatory responses of endothelial cells. In this study, we investigated the effects of high-density lipoproteins (HDL) on NET formation. HL-60-derived neutrophils were treated with phorbol 12-myristate 13-acetate (PMA) and further incubated with oxLDL and various concentrations of HDL for 2 h. NET formation was evaluated by quantifying extracellular DNA and myeloperoxidase. We found that the addition of native HDL partially decreased NET formation of neutrophils induced by oxLDL. This effect of HDL was lost when HDL was oxidized. We showed that oxidized phosphatidylcholines and lysophosphatidylcholine, which are generated in oxLDL, promoted NET formation of PMA-primed neutrophils, and NET formation by these products was completely blocked by native HDL. Furthermore, we found that an electronegative subfraction of LDL, LDL(–), which is separated from human plasma and is thought to be an in vivo oxLDL, was capable of promoting NET formation. These results suggest that plasma lipoproteins and their oxidative modifications play multiple roles in promoting NET formation, and that HDL acts as a suppressor of this response.

## 1. Introduction

Formation of neutrophil extracellular traps (NETs), considered one of the early immune responses by neutrophils, was first reported as a novel self-defense phenomenon in which neutrophils release their own chromosomal DNA as an extracellular net to trap pathogens [1]. Subsequently, NETs were also found in patients with non-infectious diseases, including cardiovascular diseases (CVDs) [2,3,4], autoimmune diseases [5], and cancer metastasis [6]. Recently, it has been revealed that NETs are deeply involved in thrombus formation in patients with severe COVID-19 [7]. NET formation can be induced by various factors, including bacteria, autoantibodies, cholesterol crystals, inflammatory cytokines, and oxidative stress [1,5,8]. It has also been reported that high mobility group box-1 protein, a type of damage-associated molecular pattern, induces NET formation via receptor for advanced glycation end-products [9]. NET formation is involved in vascular damage; however, the factors related to NET formation have not been fully elucidated.

Oxidized low-density lipoprotein (oxLDL) is known to promote atherosclerotic lesions by being taken up by macrophages via scavenger receptors, leading to foam cell formation. Plasma oxLDL levels are higher in patients with CVDs than in healthy subjects [10,11,12], and oxLDL has been detected in human atherosclerotic lesions [13]. Furthermore, oxidized phosphatidylcholines (oxPCs) and lysophosphatidylcholine (lysoPC), which are produced during low-density lipoprotein (LDL) oxidation, are present in atherosclerotic lesions [14,15]. However, only a few studies have investigated the effects of oxLDL and oxPCs on neutrophils. We previously reported that oxLDL, but not native LDL, promotes NET formation, suggesting the involvement of oxidized lipoproteins in atherosclerotic lesion progression through neutrophil responses [16].

The nature of oxLDL present in vivo has long been debated; however, recent studies have suggested that electronegative LDL(–), a subfraction of LDL separated using anion-exchange column chromatography, shows some characteristics that correspond to oxLDL [17,18,19]. An association between elevated plasma LDL(−) levels and CVDs has also been reported [20,21,22]. Interestingly, we found that oxidatively modified high-density lipoprotein (oxHDL), in addition to oxLDL, is present in the LDL(–) fraction [23]. High-density lipoprotein (HDL) is a lipoprotein with reverse cholesterol transport as well as anti-inflammatory and antioxidant effects, which eventually presents HDL with anti-atherosclerotic effects. In addition, HDL contains various lipid-metabolizing enzymes that detoxify lipid peroxides and prevent their accumulation in LDL [24]. In contrast, oxHDL is known to decrease cholesterol transport capacity and induce oxidative stress and inflammation, in addition to having a positive association with atherosclerotic lesions [25,26].

While we have demonstrated the effects of copper-oxidized LDL on NET formation in a previous study [16], the effects of lysoPC and oxPCs (which are produced in oxLDL) on NET formation remain unclear. Considering the in vivo milieu of atherogenesis, there is a dire need to understand the effects of other lipoproteins on NET formation. In this study, using HL-60-derived neutrophils, we found that HDL had suppressive effects on NET formation promoted by oxLDL, lysoPC, and oxPCs. Moreover, oxidation of HDL abolished its suppressive function, and oxHDL promoted NET formation. Finally, LDL(–) separated from human plasma that contains oxHDL and oxLDL also enhanced NET formation.

## 2. Results

### 2.1. Native HDL Suppresses NET Formation Enhanced by oxLDL

In this study, we investigated the effects of HDL on the enhancement of NET formation by oxLDL. HL-60-derived neutrophils were spread on 24-well plates coated with 0.01% poly-L-lysine and then treated with 0 or 20 nM phorbol 12-myristate 13-acetate (PMA). OxLDL was pre-incubated with 0, 20, or 40 μg/mL of native HDL at 37 °C for 4 h, following which the PMA-treated cells were cultured with the lipoprotein preparations for 2 h. Micrococcal nuclease (MNase) was added to the culture wells to digest the DNA released extracellularly, following which NET formation was evaluated in terms of the fluorescence intensity in the culture medium, using SYTOX^®^ Green (Figure 1a). The pre-incubation time was set to 4 h based on our previous observation that HDL gradually converted lysoPC in oxLDL to diacyl-PC at 37 °C by up to 4 h [27]. A preliminary experiment showed that the inhibition of DNA release by HDL was slightly more effective after preincubation at 37 °C for 4 h than for 1 h (Appendix A).

Consistent with our previous study [16], the addition of oxLDL to PMA-treated cells increased DNA release as compared to PMA treatment alone (Figure 1b). Pre-incubation of oxLDL with native HDL at ratios of 1:1 or 1:2 (oxLDL:native HDL, by amount of protein) reduced the DNA release in a dose-dependent manner, with DNA release being suppressed by 39% at the ratio of 1:2. The suppressive effect of HDL was observed when the cells were treated with oxLDL alone, without PMA. The treatment of neutrophils with oxLDL and PMA showed a synergistic effect; namely, the level of DNA released upon combined treatment with oxLDL and PMA was higher than the simple sum of those observed upon treatment with either PMA or oxLDL alone. HDL did not show a suppressive effect on PMA-treated cells in the absence of oxLDL at 20 μg/mL, while a higher dose of HDL slightly suppressed DNA-release from PMA-primed cells (Appendix A).

In order to ascertain whether oxidative modification of HDL affects its suppressive effect on NET formation, we performed a similar experiment using oxHDL instead of native HDL. Copper-mediated oxidation of HDL produced approximately 20 ng of thiobarbituric acid reactive substances per mg protein, which was one-third of that seen in the case of oxLDL (Appendix A). Since the lipid-per-protein ratio of LDL is approximately 3-fold higher than that of HDL, their oxidation levels were comparable. In addition, the amounts of triglyceride and total-cholesterol per each microgram of LDL or HDL decreased by 10% and 50%, respectively, during copper-induced oxidation (Appendix A). It is likely that lipid oxidation products may be generated during the copper oxidation process similarly in oxLDL and oxHDL. Copper-induced oxidation of HDL abolished the suppressive effect of native HDL on the oxLDL-mediated acceleration of NET formation. Notably, oxHDL had the potential to induce NET formation (Figure 1c).

Furthermore, the effects of native HDL and oxHDL on NET formation were evaluated in terms of the myeloperoxidase (MPO) released into the cell culture medium (Figure 1d,e). MPO was detected at approximately 60 kDa after PMA treatment, and the MPO release increased upon the addition of oxLDL after PMA priming. Although the oxLDL-dependent increase in MPO release was suppressed by 47% in the presence of native HDL, the oxidation of HDL canceled the HDL-mediated suppression of MPO release. Thus, the amount of MPO in the medium corresponded well with the amount of DNA released under these conditions. These results indicated that native HDL can suppress the NET formation promoted by oxLDL.

### 2.2. LDL(–), the In Vivo oxLDL, Promotes NET Formation

Since the coexistence of native HDL or oxHDL with oxLDL affected NET formation, we next examined LDL(–), a negatively charged LDL subfraction in human plasma that is considered the oxLDL in vivo, in order to investigate the effect of modified lipoproteins on NET formation. We previously demonstrated that the amounts of oxidative products of phospholipids in LDL(–) were less than those in copper-induced oxLDL; however, LDL(–) contained HDL particles enriched with oxidative modification of apolipoprotein A-I [23]. Native LDL, oxLDL, and LDL(–) were examined for their effect on NETs-DNA release from HL-60-derived neutrophils after treatment with or without 20 nM PMA. Although DNA release was not induced by LDL(–) itself without PMA, LDL(–) significantly enhanced PMA-induced NETs-DNA release. Native LDL did not exhibit this enhancement effect. The DNA release in the presence of LDL(–) was 33% lower than that in the presence of copper-induced oxLDL (*p* = 0.0571) (Figure 2a).

Next, MPO released from the cells into the culture medium was evaluated using Western blot (Figure 2b,c). The MPO released into the medium upon the addition of LDL(–) was significantly increased by more than 2-fold as compared to that observed upon PMA treatment alone, while it was less than that observed upon addition of oxLDL. These results indicate that LDL(–) has the ability to promote NET formation induced by PMA treatment, although the effect of LDL(–) was slightly milder than that of copper-induced oxLDL. It is likely that the difference in the oxidation products generated by oxLDL and LDL(–) may correlate with the difference in their NET-promoting abilities.

### 2.3. LysoPC and oxPC Promote NET Formation

It has been shown that copper-induced oxLDL contains more lysoPC and oxPCs than native LDL [28]. Thus, we examined its effects on the promotion of NET formation in this study. HL-60-derived neutrophils were spread on 24-well plates and treated with or without 20 nM PMA to induce NET formation, following which the cells were incubated with 10 μM 1-palmitoyl-lysoPC. As shown in Figure 3b, NET formation of PMA-primed cells increased by more than 3-fold upon the addition of lysoPC. Under the same experimental conditions, 10 μM 1-palmitoyl-2-(5-oxovaleroyl)-phosphatidylcholine (POVPC), a major oxPC species produced in oxLDL, with a short-chain aldehyde group at the *sn-*2 position (Appendix A), promoted the NET formation of PMA-primed cells (Figure 3c).

The effect of oxPC on NET formation was not limited to POVPC. POVPC and 1-palmitoyl-2-glutaroyl PC (PGPC) are oxPCs derived from 1-palmitoyl-2-arachidonoyl-phosphatidylcholine (PAPC), which contains 5-carbon short chains, while 1-palmitoyl-2-azelaoyl phosphatidylcholine (PAzPC) is another oxPC produced from 1-palmitoyl-2-linoleoyl PC (PLPC), which contains a 9-carbon short chain (Appendix A). We also examined whether these lipids could promote NET formation. In these experiments, the cells were treated with 50 nM PMA so that the levels of DNA release were slightly higher than those in the other experiments. PAPC did not show any additive effect on NET formation, and its oxidized products (POVPC and PGPC) accelerated NET formation. PLPC showed an increase in NET formation, while PAzPC showed an even stronger effect on cells (Appendix A). Overall, oxPCs significantly enhanced DNA release as compared to unoxidized diacyl-PCs. These results suggest that lysoPC and various oxPCs can promote NET formation induced by PMA.

### 2.4. HDL Suppresses NET Formation Promoted by Phospholipids

Next, we examined the impact of native HDL on NET formation of HL-60-derived neutrophils primed with PMA, lysoPC, or various oxPCs. LysoPC and oxPCs were pre-incubated with HDL, for 4 h at 37 °C, and then added to the PMA-treated cells (Figure 3a). DNA released into the culture supernatant was measured using the same procedure as previously described. Although there were differences in the degree of DNA release between lysoPC and oxPCs treatments, pre-incubation of these phospholipids with native HDL canceled the lipid-mediated enhancement of DNA release to the level observed in case of PMA treatment alone (Figure 3d). These results suggested that lysoPC and oxPCs are at least partly responsible for the NET-promoting effects of oxLDL. In contrast, mixing lysoPC or POVPC with oxHDL showed that oxHDL had no suppressive effect on NET formation (Figure 3b,c). Moreover, oxHDL alone accelerated the NET formation induced by lysoPC and POVPC. Altogether, these findings indicate that native HDL is capable of suppressing NET formation that is enhanced by oxLDL, lysoPC, and oxPCs.

### 2.5. Lecithin Cholesterol Acyltransferase (LCAT) Is Not Involved in the Inhibitory Effects of HDL on NET Formation Enhanced by oxLDL and Oxidized Phospholipids

To further investigate the ability of HDL to inhibit NET formation promoted by oxLDL, lysoPC, and oxPCs, we confirmed the involvement of LCAT, a lipid-metabolizing enzyme. LCAT on HDL catalyzes the acylation of free cholesterol and other various substrates, including lysoPC to produce diacyl-PC [27]. To investigate whether the inhibitory effect of HDL was mediated by LCAT activity, native HDL was pre-incubated with 5,5′-dithio-bis-(2-nitrobenzoic acid) (DTNB), a blocking reagent for thiol groups, and then mixed with oxLDL, lysoPC, or POVPC before addition to the cells (Figure 4a). The addition of DTNB did not change the suppressive effect of HDL on the enhanced NET formation by oxLDL (Figure 4b). Similar results were observed when native HDL treated with DTNB was examined in combination with lysoPC or POVPC (Figure 4c,d). These results suggest that the function of HDL in suppressing NET formation promoted by oxLDL, lysoPC, or oxPC is mediated by LCAT-independent mechanisms.

## 3. Discussion

In this study, we investigated the effects of HDL on the NET formation of HL-60-derived neutrophils and found that native HDL acted as a suppressor of the enhancement of NET formation by oxLDL as well as by lysoPC or oxPCs. We also confirmed that copper-induced oxidation deprived native HDL of its suppressive effect and that the resulting oxHDL promoted NET formation. LysoPC and oxPCs, which were generated from LDL and HDL during oxidation, promoted NET formation that was blocked by HDL. Furthermore, LDL(–), the in vivo oxLDL fraction isolated from human plasma, was found to have the potential to promote NET formation. This study provides the first evidence that these lipoproteins and their oxidative conditions in combination have interactive effects on the enhancement of NET formation.

### 3.1. Suppressive Effect of HDL on NET Formation

We investigated the role of lipoproteins in the acceleration of NET formation, which is a novel mechanism related to CVDs. Since HDL is well-known as an anti-atherogenic component in circulation, we examined the effect of HDL on PMA-induced NET formation in the presence of oxLDL or its oxidized phospholipid products. We demonstrated that native HDL partially suppressed the DNA release promoted by oxLDL (Figure 1), whereas HDL completely abolished the DNA release enhanced by lysoPC and oxPCs (Figure 3). It is speculated that either of the phospholipid products (lysoPC and oxPCs) or oxidatively modified proteins in oxLDL could accelerate NET formation of neutrophils, and it is likely that HDL acted on oxPCs during pre-incubation in a neutrophil-independent manner. Emert et al., reported that HDL repressed the transcriptional function of oxPCs in endothelial cells, which required the direct interaction of HDL with oxPCs [29]. Since the oxPCs used in this study are less hydrophobic than diacyl-PCs because their *sn*-2 acyl chains are substituted with short-hydrophilic moieties, they have been shown to move easily from oxLDL particles to other lipoprotein particles or cellular membranes [27]. Furthermore, oxPCs can covalently bind to apolipoprotein A-I (apoA-I) [30,31,32]. Several studies have reported that oxPCs with γ-hydroxy-alkenal or aldehyde functions form Michael adducts with histidine and cysteine residues of apoA-I [30,31,32], and certain oxPCs bind more readily to apoA-I than to plasma proteins [31]. Therefore, it is likely that oxPCs translocate from oxLDL to HDL before acting on neutrophils and can be trapped in HDL particles by hydrophobic interactions or covalent binding. The transfer of oxPCs to neutrophils could depend on the ratio of free oxPCs to lipoprotein particles, and this would be one reason to explain why the addition of oxHDL to oxLDL increased NET formation while oxHDL showed little effect on lysoPC- or POVPC-induced DNA release.

ApoA-I plays an essential role in the structural integrity and antioxidant properties of HDL [33]. In addition, apoA-I is involved in lipid metabolism by sustaining LCAT enzymatic reactions. Since LCAT catalyzes the acylation of lysoPC to produce diacyl-PC [23], we examined whether LCAT activity is involved in the suppressive effect of HDL on NET formation. Our data showed that DTNB did not alter the suppressive effect of HDL on NET formation, thereby suggesting that LCAT did not play a causative role in the effect of HDL on the suppression of NET formation. One possible explanation is that transacylation of lysoPC by LCAT to form diacyl-PC concomitantly generates new de-acylated lipids, which would cancel the acylation of lysoPC.

### 3.2. Activity of oxPCs

We demonstrated that several short chain-containing oxPCs accelerated NET formation of PMA-primed neutrophils. These short-chain oxPCs are produced in oxLDL and atherosclerotic lesions [15,34]. In addition, lysoPC is a common constituent of LDL, but accumulates when LDL is oxidized. We decided the concentrations of oxPC or lysoPC to be tested in this study based on our previous study, which reported the acceleration of NET formation upon treatment of PMA-stimulated neutrophils with 20 μg/mL oxLDL [16]. Since LDL contains almost equal amounts of proteins and phospholipids by weight and the molecular weights of PCs range from 700–850, more than 25 μM PCs are estimated to be present in 20 μg/mL oxLDL. Since as much as 33% of the phospholipids in copper-induced oxLDL are composed of oxPCs and lysoPC [28], we assumed that the oxidized products of PCs in 20 μg/mL oxLDL are slightly less than 10 μM. In the literature, it has been reported that oxPAPC (a mixture of oxidative derivatives from PAPC) induced NET formation [35,36]; however, the amount of DNA released was evaluated upon addition of as much as 200 μg/mL oxPAPC, which is about 245 μM. In our study, we demonstrated acceleration of NETs formation by low concentrations of oxPCs (10 μM).

We tested several short-chain oxPCs with different *sn*-2 acyl chains, together with lysoPC, and found that all of them promoted NET formation. Although POVPC contains 5-carbon aldehyde at the *sn*-2 position, it has often been reported as a strong inducer of inflammatory responses [37,38]. PAzPC, a 9-carbon carboxy group at *sn-*2, induced NET formation more effectively than POVPC. These results suggest that the length of the oxidized acyl chain at the *sn*-2 position, rather than the functional group in the oxidized portion, may contribute to NET formation.

### 3.3. LDL(–) Is Capable of Promoting NETs Formation

LDL(–), an electronegative subfraction of LDL separated by means of anion-exchange chromatography, has been studied for many years and is thought to be a candidate for in vivo oxLDL [20]. We found that the amount of LDL(–) in the plasma of patients with acute myocardial infarction was 3-times higher than that in normal subjects [23]. LDL(–) is characterized by the presence of HDL particles with extensive modification of apoA-I, which directly corresponds to its electronegativity. We discovered that LDL(–) promoted NET formation of PMA-primed neutrophils, but it did not initiate significant NET formation (Figure 2a). While LDL(–) is enriched with oxidative modifications in apolipoproteins of LDL and HDL [23], it contains fewer oxidized lipids than copper-induced oxLDL [39]. These properties of LDL(–) might explain the milder action of LDL(–) on the acceleration of NET formation as compared to that of oxLDL. This observation is critical evidence that supports the involvement of lipoproteins in NET-related diseases in vivo. Given that NETs are formed in various pathological conditions/diseases [5,40,41,42] and oxidized lipoproteins accumulate in the circulation or various lesions [25,43,44,45,46], it is likely that oxidized lipoproteins are involved in disease progression through NET formation.

### 3.4. Overall implication of lipoproteins in NET formation

As NET formation is involved in atherosclerotic lesion formation [2,3], we considered the factors affecting NET formation in the blood vessel environment. In our previous report, we found that oxLDL promoted NET formation in vitro and that NETs enhanced the inflammatory response of vascular endothelial cells [16]. This led us to speculate that the suppression of NET formation may contribute to the suppression of CVDs. In the present study, we found that native HDL attenuated the enhancement of NET formation by oxLDL and its lipid components. However, the suppressive effect of HDL was lost when it was oxidized. These results suggest that circulating HDL could prevent the induction of NET formation by absorbing oxidized lipids. We also observed that LDL(−) mildly enhanced NET formation. Taken together, both LDL and HDL in their native and oxidized forms play multiple roles in NET formation. Further studies are necessary to elucidate the involvement of these lipoproteins in NET-related diseases in vivo.

## 4. Materials and Methods

### 4.1. Materials

All-*trans* retinoic acid (AtRA), PMA, and DTNB were purchased from Fujifilm Wako Pure Chemical Co. (Osaka, Japan). Poly-L-lysine solution (P4707), 1-palmitoyl-lysophosphatidylcholine (16:0-LysoPC), and PAPC were purchased from Sigma-Aldrich (St. Louis, MO, USA). Other phospholipids, including POVPC, PGPC, and PLPC were purchased from Avanti Polar Lipids Inc. (Alabaster, AL, USA). PAzPC was purchased from Cayman Chemical Company (Ann Arbor, MI, USA). MNase was purchased from TAKARA Bio Inc. (Shiga, Japan). SYTOX^®^ Green was purchased from Thermo Fisher Scientific (Waltham, MA, USA). Anti-human MPO antibody was purchased from Dako (A0398; Carpinteria, CA, USA). A horseradish peroxidase-conjugated anti-rabbit IgG was purchased from GE Healthcare (catalog no. NA934; Buckinghamshire, UK).

### 4.2. Culture of HL-60 Cells and Differentiation into Neutrophil-Like Cells

HL-60 cells, a human promyelocytic leukemia cell line, was purchased from American Type Culture Collection (Manassas, VA, USA), and the cells were cultured in RPMI-1640 medium supplemented with 5% deactivated (for 30 min at 56 °C) fetal bovine serum (1202144; Gibco, Waltham, MA, USA), 50 U/mL penicillin, and 50 μg/mL streptomycin (15140-122; Gibco) in 10 cm dishes (430591; Corning, Glendale, AZ, USA), with a change of medium every 3–4 days. To differentiate the cells into neutrophil-like cells, 2.0 × 10^6^ cells/dish were stimulated with 2 μM AtRA for 4 days, as reported previously [16].

### 4.3. Preparation of Phospholipids

Aliquots of lysoPC, PAPC, POVPC, PGPC, PLPC, or PAzPC were placed in sample tubes, dried under nitrogen gas, and suspended in serum-free RPMI-1640 (phenol red free) medium by vortex mixing for approximately 30 s, following which the cells were treated with them in an ultrasonic bath for 10 min at room temperature.

### 4.4. Fractionation of LDL and HDL from Human Plasma, by Means of Ultracentrifugation

This study was approved by the Ethics Committee of Showa University (approval no. 231). Written informed consent was obtained in accordance with the Declaration of Helsinki, and all human subjects voluntarily gave their consent for participation in this study. Fractionation of LDL and HDL was performed as reported previously [16,23]. Whole blood (50 mL) from healthy subjects was centrifuged twice at 200× *g* and 4 °C for 15 min in order to obtain plasma, which was supplemented with 250 μM ethylenediaminetetraacetic acid (EDTA) (pH 7.4) to avoid oxidation. Phosphate-buffered saline containing 250 μM EDTA (E-PBS, 0.9 mL) was layered on top of 2.7 mL of the plasma in 4PC tubes (Hitachi High-Technologies Co., Tokyo, Japan), following which the tubes were centrifuged at 500,000× *g* and 4 °C for 7 min to remove chylomicrons. Again, 0.9 mL of E-PBS was layered on the remaining samples, and very low-density lipoprotein was removed by means of centrifugation at 500,000× *g* and 4 °C, for 2.5 h. The remaining lower layer was mixed with 0.54 mL 0.5 g/mL KBr, to adjust the density to 1.063. After centrifugation at 500,000× *g* and 4 °C for 2.5 h, the LDL floating up to the surface was collected and dialyzed against E-PBS. The remaining lower layer was mixed with 0.5 g/mL KBr, to adjust the density to 1.210. After centrifugation at 500,000× *g* and 4 °C for 4 h, the HDL floating up to the surface was collected and dialyzed against E-PBS. The isolated LDL and HDL were stored at 4 °C.

### 4.5. Separation of LDL(–) using Anion-Exchange Chromatography

Separation of LDL(–), an electronegative LDL subfraction, was performed as previously reported with slight modifications [20,23]. Native LDL was passed through a desalting column (HiTrap™ Desalting; GE Healthcare, Chicago, IL, USA) using buffer A [10 mM Tris-HCl/1 mM EDTA (pH 7.4)]. Approximately 6 mg native LDL was placed on a HiTrap™ Q Fast Flow column (17-5053-01, GE Healthcare, Chicago, IL, USA) followed by stepwise elution using 0 to 1.0 M NaCl. The peak of absorbance at the wavelength of 280 nm eluted with 0.5 M NaCl was collected as the LDL(–) fraction. The recovered LDL(–) fractions were gently mixed with 0.5 mg/mL KBr to adjust the density to 1.063 and then centrifuged at 500,000× *g* for 2.5 h. The floating LDL(–) fraction was collected, dialyzed against E-PBS, and stored at 4 °C.

### 4.6. Oxidation of LDL and HDL

The EDTA in the LDL or HDL preparations was removed using a HiTrap™ Desalting column (GE Healthcare, Chicago, IL, USA), following which the protein concentration was determined using bicinchoninic acid protein assay reagent (Pierce, Thermo Fisher Scientific, Waltham, MA, USA). Two milliliters of 0.2 mg/mL LDL or HDL were placed in a round-bottom test tube and the oxidation reaction was started by adding 5 μM CuSO_4_. After the sample was incubated at 37 °C for 3 h with gentle shaking, the mixture was placed on ice and 250 μM EDTA was added to stop the reaction. The sample volume was adjusted to 2 mL to compensate for the water that evaporated during oxidation.

Thiobarbituric acid-reactive substances were measured as previously described [47], with slight modifications. TBA reagent was prepared by dissolving 15 mg of 2-thiobarbituric acid in 4 mL of 15% trichloroacetic acid with 0.25 N hydrochloric acid while heating. TBA reagent (0.4 mL) was added to 0.2 mL lipoprotein samples and the mixture was heated at 100 °C for 15 min. After the samples were cooled in water, they were centrifuged at 750× *g* for 5 min, and the absorbance of the supernatants was measured at the wavelength of 540 nm.

### 4.7. Stimulation of HL-60-Derived Neutrophils

Poly-L-lysine-coated plates were prepared by incubating 0.01% poly-L-lysine solution in 24-well plates (SIAL0526; Sigma, St. Louis, MO, USA and 3526, Costar^®^, Corning, Glendale, AZ, USA), for 5 min at room temperature. The HL-60-derived neutrophil-like cells were collected and washed once with serum-free RPMI-1640 (phenol red free) medium. The cell suspension was seeded in 24-well plates (5.0 × 10^5^ cells/well) and cultured for 30 min. NETs were induced by treatment with 20 nM PMA, at 37 °C under 5% CO_2_ for 30 min, followed by an additional 2 h incubation with oxLDL or the various other PCs. When HDL was used for co-incubation with oxLDL and/or PCs, it was pre-incubated with lysoPC, oxPC, or oxLDL in round-bottom tubes filled with argon gas for 4 h at 37 °C, with gentle shaking. To inhibit LCAT activity, 1 mg/mL HDL was mixed with 1 mM DTNB in round-bottom tubes filled with argon gas for 30 min at 37 °C with gentle shaking before pre-incubation with oxLDL or PCs, as described in [27].

### 4.8. Fluorometric Quantitation of NETs-DNA

Quantification of NETs-DNA was performed using a slightly modified version of a previous report [16]. After stimulating the cells, the samples were treated with 1 U/mL MNase to degrade the extracellular DNA released upon NET formation. The culture medium from each sample was recovered by means of centrifugation at 1800× *g* for 10 min. The obtained supernatant was then transferred to 96-well plates and mixed with 1 μM SYTOX^®^ Green. Fluorescence (ex: 485 nm, em: 525 nm) was measured using Varioskan^®^ Flash (Thermo Fisher Scientific, Waltham, MA, USA).

### 4.9. Detection of MPO Released upon NET Formation

Culture medium samples obtained from the HL-60-derived neutrophils after stimulation were subjected to sodium dodecyl sulfate-polyacrylamide gel electrophoresis on a 10% polyacrylamide gel and then transferred to a polyvinylidene fluoride membrane. The membrane was immersed in a blocking solution [2% skim milk (190-12865, Fujifilm Wako Pure Chemical Co., Osaka, Japan)/tris-buffered saline containing 0.1% Tween 20 (TTBS)] with shaking for 1 h. The membrane was treated with an anti-MPO antibody diluted in a ratio of 1:10,000 in 2% skim milk dissolved in TTBS for 30 min at room temperature, with shaking, and then again overnight at 4 °C. Thereafter, the membrane was washed with TTBS and treated with an anti-rabbit IgG antibody, diluted in the ratio of 1:3000 in 0.5% skim milk/TTBS, for 2 h at room temperature. The obtained bands were then detected using chemiluminescence reagent [ECL Prime Western Blotting Detection Reagent (RPN2232, Cytiva, Waltham, MA, USA)] and ImageQuant™ LAS 500 (GE Healthcare).

### 4.10. Data Analysis

Data analysis was performed using JMP^®^ Pro 16.0.0 software. The normal distribution of the values was evaluated using the Shapiro–Wilk test, equal variance was confirmed using the O’Brien test, and statistical significance was calculated using analysis of variance. The Tukey–Kramer’s test was conducted to examine the differences, while the Wilcoxon test was used for samples that were not normally distributed. *p*-values less than 0.05 were considered as statistically significant.

## Figures and Tables

**Figure 1 ijms-23-13992-f001:**
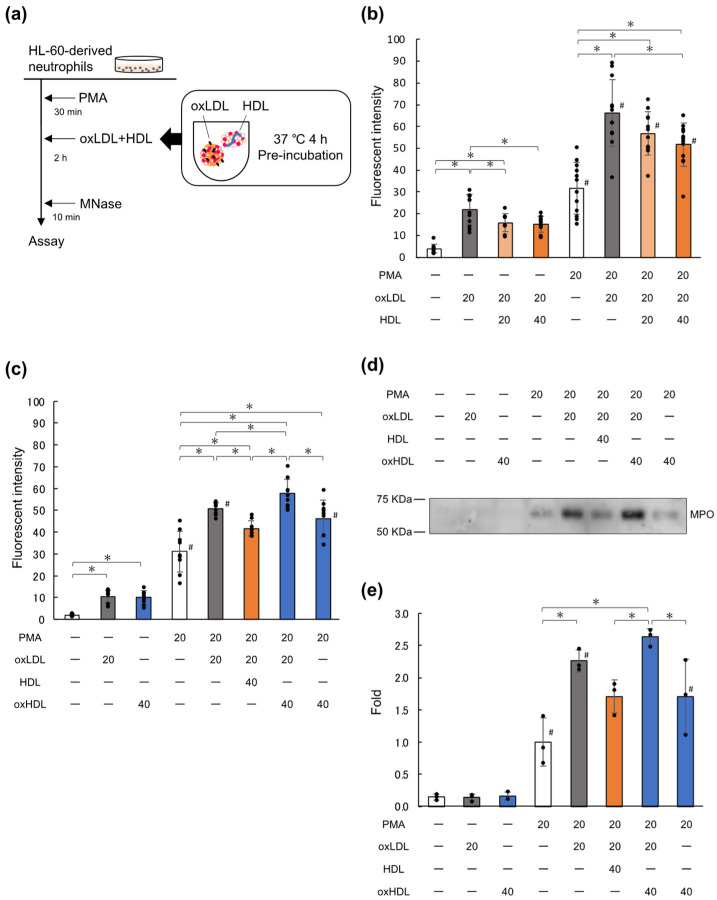
HDL attenuates the oxLDL-enhanced NET formation of HL-60-derived neutrophils. (**a**) Experimental scheme. OxLDL was pre-incubated with native HDL or oxHDL at 37 °C for 4 h, before being added to the cells. HL-60 cells were differentiated into neutrophil-like cells using 2 μM AtRA treated with or without 20 nM PMA for 30 min, and then treated with lipoproteins for 2 h. The medium was treated with 1 U/mL MNase for 10 min in order to cleave the DNA into short segments. The fluorescence intensity of the medium supernatant was then measured using 1 μM SYTOX^®^ Green. (**b**) Effect of native HDL on oxLDL-mediated NET formation. OxLDL (20 μg/mL) was pre-incubated without HDL or with 20 or 40 μg/mL HDL for 4 h, following which the lipoproteins were added to the neutrophils for 2 h. The data represent the mean ± SD of four experiments, each carried out in triplicate; each dot represents one data point. (**c**) Effect of oxHDL on oxLDL-mediated NET formation. Cells primed using PMA were incubated with 20 μg/mL oxLDL in the presence of 40 μg/mL native HDL or oxHDL for 2 h. In addition, some cells were incubated with oxHDL alone, for 2 h. The data represent the mean ± SD from three experiments, each carried out in triplicate. (**d**,**e**) The MPO released in the medium was detected using Western blot in aliquots of the culture supernatant obtained in (**c**). A representative Western blot (**d**) and its quantification using ImageJ software (**e**) have been shown. The data are shown as the relative value to the band intensity of the control sample (no lipoprotein after PMA treatment). The data from three runs are shown. Asterisks indicate statistical significance (* *p* < 0.05), while sharps indicate statistical significance against the corresponding conditions without PMA (# *p* < 0.05). HDL, high-density lipoprotein; NETs, neutrophil extracellular traps; oxLDL, copper-oxidized low-density lipoprotein; oxHDL, copper-oxidized high-density lipoprotein; AtRA, all-*trans* retinoic acid; PMA, phorbol 12-myristate 13-acetate; MNase, micrococcal nuclease; MPO, myeloperoxidase.

**Figure 2 ijms-23-13992-f002:**
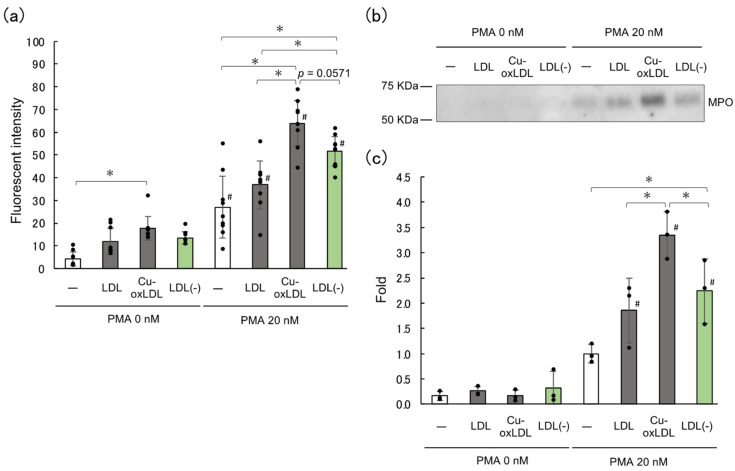
LDL(–) enhances the NET formation of HL-60-derived neutrophils. LDL(–), an electronegative subfraction of LDL, is a fraction that enriches oxLDL in vivo. (**a**) Fluorometric quantification of DNA release. HL-60 cells were differentiated into neutrophil-like cells using 2 μM AtRA and treated with or without PMA for 30 min. Following that, native LDL, Cu-oxLDL or LDL(–) (final concentration of 20 μg/mL) was added to and incubated with the cells for 2 h. The DNA released into the medium was digested with MNase for 10 min, and the DNA fragments thus obtained were quantified fluorometrically using SYTOX^®^ Green. The data represent the mean ± SD from three experiments, each carried out in triplicate; each dot represents one datapoint. (**b**,**c**) MPO in the medium was detected in the culture supernatant prepared as described in (**a**) using Western blot. Representative data from three runs are shown. Band intensity was calculated using ImageJ software (**c**). The data from the three runs are shown. Asterisks indicate statistical significance (* *p* < 0.05), while sharps indicate statistical significance compared to the corresponding conditions without PMA (# *p* < 0.05). LDL, low-density lipoprotein; Cu-oxLDL, copper-oxidized low-density lipoprotein; LDL(–), electronegative low-density lipoprotein; AtRA, all-*trans* retinoic acid; PMA, phorbol 12-myristate 13-acetate; MNase, micrococcal nuclease; MPO, myeloperoxidase.

**Figure 3 ijms-23-13992-f003:**
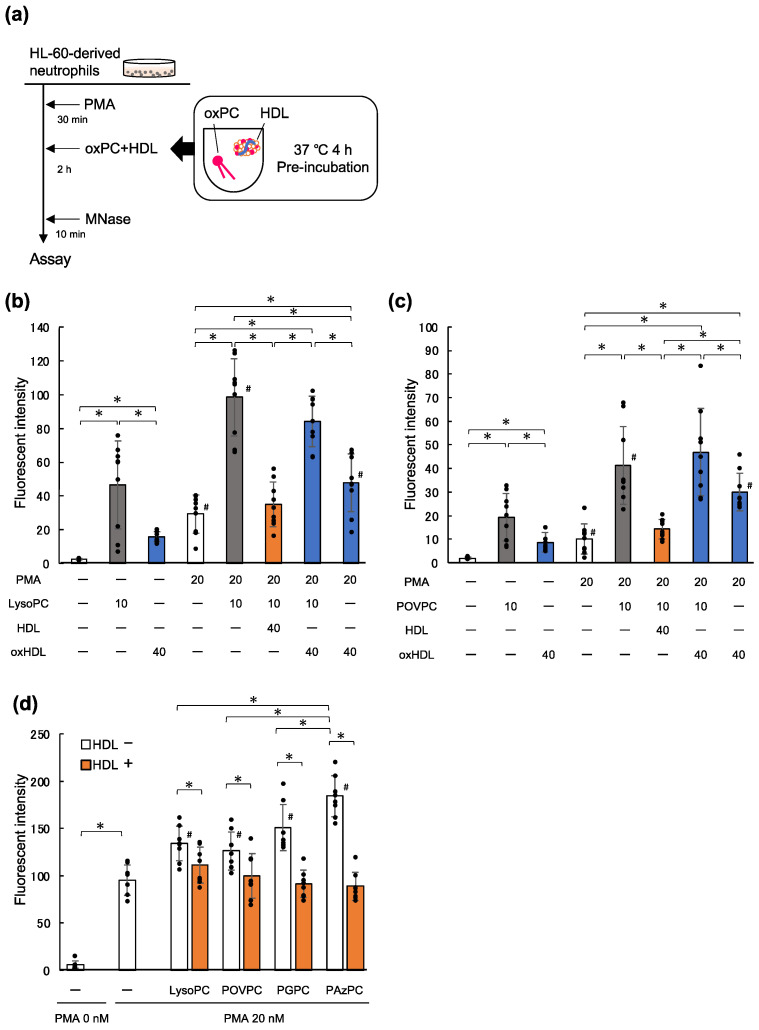
HDL inhibits enhancement of NET formation by lysoPC and oxPCs. (**a**) Experimental scheme. LysoPC or oxPCs were pre-incubated with or without 40 μg/mL HDL or oxHDL at 37 °C for 4 h. (**b**,**c**) Either 10 μM lysoPC (**b**) or POVPC (**c**) were added to the PMA-primed neutrophils in order to enhance NET formation. The data represent the mean ± SD from four experiments, each carried out in triplicate; each dot represents one datapoint. Asterisks indicate statistical significance (* *p* < 0.05), while sharps indicate statistical significance against the corresponding conditions without PMA (# *p* < 0.05). (**d**) Similarly, HL-60 derived neutrophil-like cells were stimulated with PMA. Following that, each phospholipid (lysoPC, POVPC, PGPC, or PAzPC; final concentration of 10 μM), with or without 40 μg/mL native HDL, was added to the cells, and the cells were incubated with them for 2 h. The data represent the mean ± SD from four experiments, each carried out with duplicates; each dot represents one datapoint. Asterisks indicate statistical significance (* *p* < 0.05), while sharps indicate statistical significance against cells treated with 20 nM PMA alone (# *p* < 0.05). HDL, high-density lipoprotein; NET, neutrophil extracellular trap; lysoPC, lysophosphatidylcholine; oxPCs, oxidized phosphatidylcholines; oxHDL, oxidized high-density lipoprotein; POVPC, 1-palmitoyl-2-(5-oxovaleroyl)-phosphatidylcholine; PMA, phorbol 12-myristate 13-acetate; PGPC, 1-palmitoyl-2-glutaryl-phosphatidyl-choline; PAzPC, 1-palmitoyl-2-azelaoyl phosphatidylcholine.

**Figure 4 ijms-23-13992-f004:**
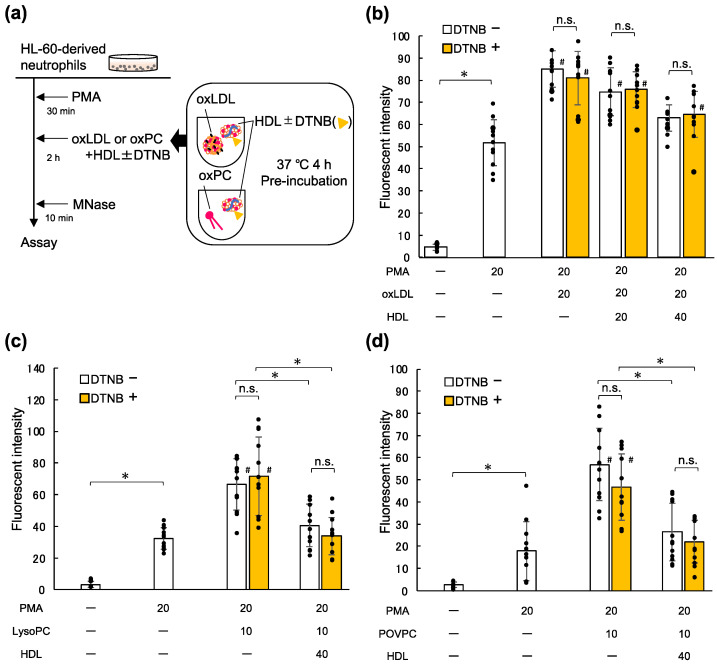
HDL suppresses NET formation even when LCAT is inhibited with DTNB. (**a**) Experimental scheme. HDL was pre-treated with or without 1 mM DTNB, an inhibitor of LCAT activity, at 37 °C for 30 min, and then further incubated with oxLDL at 37 °C for 4 h. HL-60 cells were differentiated into neutrophil-like cells using 2 μM AtRA and then treated with or without PMA for 30 min, following which the cells were incubated with 20 μg/mL oxLDL alone or with either 20 or 40 μg/mL DTNB-treated HDL for 2 h. After incubation, the DNA released upon MNase digestion was stained with SYTOX^®^ Green. (**b**) HL-60-derived neutrophils primed with PMA were incubated for 2 h with 20 μg/mL oxLDL or oxLDL plus HDL pre-treated with 1 mM DTNB. The data represent the mean ± SD from four experiments, each carried out in triplicate; each dot represents one datapoint. Asterisks indicate statistical significance (* *p* < 0.05), while sharps indicate statistical significance against the cells treated with 20 nM PMA alone (# *p* < 0.05). (**c**,**d**) HL-60-derived neutrophil-like cells were treated with or without PMA, then incubated for 2 h with 10 μM lysoPC alone or that plus 40 μg/mL HDL pre-treated with or without DTNB (**c**). Similarly, cells treated with PMA were incubated for 2 h with 10 μM POVPC alone or that plus 40 μg/mL native HDL, in the presence or absence of DTNB (**d**). The data represent the mean ± SD from four experiments, each carried out in triplicate; each dot represents one datapoint. Asterisks indicate statistical significance (* *p* < 0.05), while sharps indicate statistical significance against the cells treated with 20 nM PMA alone (# *p* < 0.05). HDL, high-density lipoprotein; NET, neutrophil extracellular trap; LCAT, lecithin cholesterol acyltransferase; DTNB, 5,5′-dithio-bis-(2-nitrobenzoic acid); oxLDL, oxidized low-density lipoprotein; AtRA, all-*trans* retinoic acid; PMA, phorbol 12-myristate 13-acetate; lysoPC, lysophosphatidylcholine; POVPC, 1-palmitoyl-2-(5-oxovaleroyl)-phosphatidylcholine; n.s., not significant.

## Data Availability

Not applicable.

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
