# Peer review of "High-Density Lipoprotein Suppresses Neutrophil Extracellular Traps Enhanced by Oxidized Low-Density Lipoprotein or Oxidized Phospholipids"

_ijms, 2022, doi:10.3390/ijms232213992_

Round 1

Reviewer 1 Report

Authors investigated the effects of high-density lipoproteins (HDL) on the formation of neutrophil extracellular traps (NETs) of phorbol myristate acetate (PMA)-primed HL-60-derived neutrophils induced by copper-oxidized low-densitiy lipoproteins. HDL, but not oxidized HDL, partially decreased NET formation of neutrophils induced by oxLDL and also by oxidized phosphatidylcholines and lysophosphatidylcholine. Furthermore, an electronegative subfraction of LDL, separated from human plasma, induces NET formation.  Authors concluded from these results that plasma lipoproteins and their oxidative modifications play a role in NET formation and that HDL might act to suppress this response.

The manuscript is well written and presents interesting findings. However, some additional informations would increase the value of these data. 

Major points:

The composition of LDL and oxLDL should be defined in more detail, especially the lipid and protein composition. To measure the lipid composition special methods should be used, for example liquid chromatography.

It would also be helpful to measure the lipid composition of HDL and oxHDL to understand why oxHDL did not shown any effect on oxLDL-induced NET formation.

NET formation and the effect of HDL should also be confirmed by additional methods such as immunhistochemistry by using antibodies such as those against citrullinated histone.

HDL is always preincubated with oxLDL or oxphospholipids, which is not explained in more detail. What is the effect without any preincubation?

The effect of HDL on oxLDL-induced NET formation should be confirmed by using human peripheral blood neutrophils

Minor points:

In the figures, the effects of HDL alone are always missing.

Description of effects of oxHDL on primed neutrophils is missing.

Author Response

We appreciate the editor and reviewers for taking the time to provide positive and constructive comments. We have added a few additional data and revised a part of results and discussion sections to respond to the comments. Following is the point-by-point answers to the reviewing comments.

Major points:

#1 The composition of LDL and oxLDL should be defined in more detail, especially the lipid and protein composition. To measure the lipid composition special methods should be used, for example liquid chromatography.

It would also be helpful to measure the lipid composition of HDL and oxHDL to understand why oxHDL did not shown any effect on oxLDL-induced NET formation.

Thank you for pointing out the alteration of lipid composition of LDL and HDL after oxidation which we did not mention in the original manuscript. During the oxidation of lipoproteins, not only PCs but neutral lipids (TG, cholesterol and cholesterol ester) are oxidized. Some lipids are converted to small hydrophilic derivatives that are not recovered by organic solvents. Therefore, it is very difficult to quantify all the products in oxidized lipoproteins.

To answer to the reviewer’s question at least in part, we measured the amount of the TG and total-cholesterol each lipoprotein, using enzymatic assay kits (triglyceride E-test kit Wako and cholesterol E-test kit Wako), since extraction of lipids is not required by these methods. The results are shown in Supplementary Table S1. One μg protein of LDL contains 0.22 μg of TG and 1.46 ug of total cholesterol (T-Chol), while 1 μg of HDL contains 0.09 μg of TG and 0.39 μg of T-Chol, which correspond well to the normal lipoprotein compositions found in the literature. On the other hand, copper oxidation reduced the amount of TG by 10% and T-Cho by 50% both in LDL and HDL. It is highly probable that loss of phospholipids during oxidation of the lipoproteins occurs similarly to those of neutral lipids. We revised the Results section to explain the results shown in Table S1 and the above information (see Section 2.1., lines 129-133).

The reviewer questioned the reason for the apparent difference of oxHDL effects on NET formation of HL-60-derived neutrophils. Namely, oxHDL increased NET formation by oxLDL (Figure1 c,e), whereas it did not change NET formation accelerated by lysoPC or oxPCs (Figure3 b,c). We speculate that the difference may be explained by the absence or presence of lipoproteins in the assay conditions. oxLDL or oxHDL contains various lipids, oxidized lipids and proteins. Many components are packaged in those lipoprotein particles, but some relatively hydrophilic lipid products could be equilibrated between the particles and the aqueous phase. Mixing oxHDL with oxLDL would not change the ratio of lipoprotein particles and free hydrophilic lipids so much and increase the total concentration of oxidized lipid. On the other hand, addition of oxHDL to lysoPC (or oxPCs) would change the ratio of lipoprotein particles and free hydrophilic lipids so that equilibration of lysoPC or oxPCs to neutrophils may be attenuated in the presence of oxHDL particles. We revised the Discussion section to include the above information (see Section 3.1., lines 309-312).

#2 NET formation and the effect of HDL should also be confirmed by additional methods such as immunohistochemistry by using antibodies such as those against citrullinated histone.

We agree with the reviewer that citrullination of histones is an important point as a NETs marker. We have tried detecting citrullinated histone, however, we could not so far obtain a clear signal either by immunostaining or by western blotting. Since we failed to detect the positive-control samples under our current conditions, we consider the antibody we used may have a problem. As mentioned in the previous study [1], immunodetection of citrullinated histones had shown significant inconsistency and antibodies against citrullinated histones showed lot-to-lot variability. We would like to check this point later, however, we cannot provide an additional data for it at the moment.

Reference

  1. Neeli I, Radic M. Current Challenges and Limitations in Antibody-Based Detection of Citrullinated Histones. Front Immunol. 2016,7,528. doi: 10.3389/fimmu.2016.00528.

#3 HDL is always preincubated with oxLDL or oxphospholipids, which is not explained in more detail. What is the effect without any preincubation?

We did not examine adding HDL without preincubation because it is technically impossible to add oxLDL or oxPCs and HDL at exactly the same time. To make the handling of the experimental manipulation constant, we considered preincubation of HDL with some stimulants such as oxLDL or oxPCs is a good way.

Our laboratory showed previously [Ref.23 in the manuscript] that oxLDL and oxHDL coexist in vivo oxLDL (LDL(-)). Considering the intravascular environment, we thought it would be rather natural for oxLDL and HDL to coexist. Furthermore, we reported that upon mixing HDL with oxidized LDL at 37°C, lysoPC is converted to diacyl PC by LCAT, and this reaction was significantly accelerated with longer mixing time, specifically, 4 h rather than 1 h [Ref.27 in the manuscript]. A preliminary experiment performed in the early stage of this research showed that inhibition of DNA release by HDL was slightly more effective after 4 h of preincubation at 37°C than after 1 h of preincubation. This result was added as Supplementary Figure S1. Based on these considerations, we decided to preincubate for 4 h. We revised the Results section and included the above information (see section 2.1., lines 84-88).

#4 The effect of HDL on oxLDL-induced NET formation should be confirmed by using human peripheral blood neutrophils

Thank you for pointing out an important point. We have performed an additional experiment using human neutrophils (see the attached PDF file for the reviewer’s reference). We isolated neutrophils from human peripheral blood using Percoll gradient centrifugation by the same procedure as in our previous reports [Ref.16 in the manuscript], then the cells were stimulated under the current experimental conditions except that PMA concentration was reduced to 5 nM, since human neutrophils are more sensitive than HL-60-derived neutrophils. As shown in a reference data for the editor, addition of HDL showed a tendency to reduce DNA release from human neutrophils treated with PMA and oxLDL by 30% (p=0.06), unfortunately it did not reach statistically significance.

Minor points:

#1 In the figures, the effects of HDL alone are always missing.

We added a data to show the effect of HDL alone to the PMA-primed neutrophils as Supplementary Figure S2. The data showed that the HDL alone may have a slight suppressive effect on NET formation of the cells. Therefore, we added a brief explanation in the revised manuscript. It is possible that HDL suppresses NET formation by trapping oxPCs and lysoPCs derived from oxLDL, however, HDL may also act in part as a suppressor of NET formation by reducing ROS generated in neutrophils. We revised the Results section to explain Supplementary Figure S2 (see lines 97-99).

#2 Description of effects of oxHDL on primed neutrophils is missing

Thank you for your suggestion. In our data, the effect of oxHDL on neutrophils is shown in figure1c and 3 c,e. And this result was mentioned briefly in the Results section (Section 2.1., line 124-125).

Reviewer 2 Report

The manuscript submitted by Ohinata et al describes the effect of oxidized and nonoxidized lipids on nHL60-netosis in response to PMA stimulation. A primary outcome is the finding that oxLDL promoted extracellular DNA and myeloperoxidase release and this is quantitatively diminished when nonoxidized HDL is added. There are no significant concerns regarding the technical execution of the experiments or of the statistical means of assessing the results.

The major concern with the work is its rationale and physiological relevance. As the authors are aware the main cell types affected by accumulation of oxidized lipoproteins are macrophages and endothelial cells. While neutrophils may be detected in atherosclerotic lesions and indeed may alter the pathology, their affect is likely due to indirect effects on the macrophages and endothelial cells that cause an atherosclerotic lesion. Therefore showing slight differences in NET production amongst a pure population of neutrophils (or nHL60) does not seem compelling unless it is coupled with a biological finding (see point #3 below).

The work has 4 major limitations that dampen enthusiasm:

1. Rather than using PMA as a priming/activating agent for NETosis the use of cholesterol crystals would have improved the physiological relevance and tied the findings into the work of Papayannopoulos and colleagues: Science. 2015 Jul 17;349(6245):316-20. doi: 10.1126/science.aaa8064. Epub 2015 Jul 16. Neutrophil extracellular traps license macrophages for cytokine production in atherosclerosis

2. The concentrations of lipoproteins seem entirely arbitrary. How were they decided upon? Did the authors perform titrations of each alone and several together to find the concentrations that result in the largest effect on NETosis?

3. Is the decrease in NETosis due to addition of HDL biological meaningful? Based on the prior publication, the authors have the ability to harvest NETs released into the media of the incubated neutrophils and transfer the media to observe endothelial inflammatory responses. Whereas NET release by HDL is diminished statistically, if the authors could show a biological response the significance of the experiments would be greatly increased.

4. The authors should show differences in NETosis inhibition by HDL by another means such as H4Cit staining in support of their findings in the current manuscript?

Author Response

We appreciate the editor and reviewers for taking the time to provide positive and constructive comments. We have added a few additional data and revised a part of results and discussion sections to respond to the comments. Following is the point-by-point answers to the reviewing comments.

The work has 4 major limitations that dampen enthusiasm:

#1. Rather than using PMA as a priming/activating agent for NETosis the use of cholesterol crystals would have improved the physiological relevance and tied the findings into the work of Papayannopoulos and colleagues: Science. 2015 Jul 17;349(6245):316-20. doi: 10.1126/science.aaa8064. Epub 2015 Jul 16. Neutrophil extracellular traps license macrophages for cytokine production in atherosclerosis

Thank you for your constructive suggestion. As you pointed out, PMA is a non-physiological NET inducer, and cholesterol crystal is one of the inducers of NETs possibly present in in vivo conditions. However, we did not examine cholesterol crystals to induce NETs because we consider NET formation is likely to implicate thrombus formation in surface erosion rather than lipid-rich vulnerable plaques. Libby et al. raised a hypothesis that a great portion of coronary heart events occur in lipid-poor arteries that are somewhat damaged by surface erosion [1]. Cholesterol crystal is often found in the lipid-rich necrotic core in atheromatous plaques, however, neutrophils are not the major infiltrated cells in those lesions.

  1. Libby, P.; Pasterkamp, G.; Crea, F.; Jang, I.‐K. Reassessing the mechanisms of acute coronary syndromes. Circ. Res. 2019, 124, 150–160, doi:10.1161/CIRCRESAHA.118.311098.

#2. The concentrations of lipoproteins seem entirely arbitrary. How were they decided upon? Did the authors perform titrations of each alone and several together to find the concentrations that result in the largest effect on NETosis?

Thank you for raising a point that should not be ignored. From previous reports by several investigators including ours, plasma oxLDL concentrations were estimated to range from 7 to 35 μg/ml [2-4]. Based on this estimation, we considered a reasonable concentration to be tested in this study would be 20 μg/ml. As described in the manuscript, there are some reports examining the effect of 100 μg/ml of oxLDL on NETs formation, however, we believe our experimental conditions is closer to physiological concentrations of oxLDL in vivo. This discussion was described in our previous paper [Ref.16 in the manuscript].

Reference

  1. Byfield FJ, Tikku S, Rothblat GH, Gooch KJ, Levitan I. OxLDL increases endothelial stiffness, force generation, and network formation. J Lipid Res. 2006, 47715–23. doi: 10.1194/jlr.M500439-JLR200
  2. Holvoet P, Mertens A, Verhamme P, Bogaerts K, Beyens G, Verhaeghe R, et al. Circulating oxidized LDL is a useful marker for identifying patients with coronary artery disease. Arterioscler Thromb Vasc Biol. 2001, 21, 844–8. doi: 10.1161/01.ATV.21.5.844
  3. Itabe H and Ueda M. Measurement of plasma oxidized low-density lipoprotein and its clinical implications. Atheroscler. Thromb. 2007:14, 1-11. (review) doi: 10.5551/jat.14.1

#3. Is the decrease in NETosis due to addition of HDL biological meaningful? Based on the prior publication, the authors have the ability to harvest NETs released into the media of the incubated neutrophils and transfer the media to observe endothelial inflammatory responses. Whereas NET release by HDL is diminished statistically, if the authors could show a biological response the significance of the experiments would be greatly increased.

Thank you for your advice. We consider that this study revealed the great biological importance of HDL since it is a common constituent in human circulation. In other words, HDL may partially suppress NET formation of neutrophils in physiological conditions that would avoid unnecessary vascular damages.

As your concern, we are also very interested in how the suppression of NET formation by HDL affects NETs’ effects on endothelial cells. One of the complications of this study is that neutrophils release a variety of proteins along with DNA. For example, MPO, once released, oxidizes many materials in the region including LDL, HDL and the plasma membrane phospholipids [5, and Ref.16 in the manuscript]. Presence of HDL would be a suppressor, however, oxidative modification of HDL could change it to a propagator of NET formation and inflammatory responses. Furthermore, MPO has been reported to promote inflammation in endothelial cells, making it a therapeutic target for cardiovascular disease [6]. We would like to study the issues related your question as a future research.

Reference

  1. Yotsumoto S, Muroi Y, Chiba T, Ohmura R, Yoneyama M, Magarisawa M, Dodo K, Terayama N, Sodeoka M, Aoyagi R, Arita M, Arakawa S, Shimizu S, Tanaka M. Hyperoxidation of ether-linked phospholipids accelerates neutrophil extracellular trap formation. Sci Rep. 2017, 22, 7, 16026. doi: 10.1038/s41598-017-15668-z.
  2. Maiocchi SL, Ku J, Thai T, Chan E, Rees MD, Thomas SR. Myeloperoxidase: A versatile mediator of endothelial dysfunction and therapeutic target during cardiovascular disease. Pharmacol Ther. 2021, 221, 107711. doi: 10.1016/j.pharmthera.2020.107711.

#4. The authors should show differences in NETosis inhibition by HDL by another means such as H4Cit staining in support of their findings in the current manuscript?

We agree with the reviewer that citrullination of histones is an important point as a NETs marker. We have tried detecting citrullinated histone, however, we could not so far obtain clear signals either by immunostaining or by western blotting. Since we failed to detect the positive-control samples under our current conditions, we consider the antibody we used may have a problem. As mentioned in the previous study [7], immunodetection of citrullinated histones had shown significant inconsistency and antibodies against citrullinated histones showed lot-to-lot variability. We would like to check this point later, however, we cannot provide an additional data for it at the moment.

Reference

  1. Neeli I, Radic M. Current Challenges and Limitations in Antibody-Based Detection of Citrullinated Histones. Front Immunol. 2016,7,528. doi: 10.3389/fimmu.2016.00528.

Round 2

Reviewer 1 Report

accept in present form

Reviewer 2 Report

The authors have addressed my concerns to my satisfaction